# Novel Biomarkers of Inflammation for the Management of Diabetes: Immunoglobulin-Free Light Chains

**DOI:** 10.3390/biomedicines10030666

**Published:** 2022-03-13

**Authors:** Akira Matsumori

**Affiliations:** Clinical Research Center, Kyoto Medical Center, 1-1 Fukakusa Mukaihata-cho, Fushimi-ku, Kyoto 612-8555, Japan; amat@kuhp.kyoto-u.ac.jp

**Keywords:** anti-inflammation, B cells, biomarker, diabetes, hepatitis C virus, immunoglobulin, inflammation, light chain, nuclear factor-kappa B, virus

## Abstract

Virus infection, inflammation and genetic factors are important factors in the pathogenesis of diabetes mellitus. The nuclear factor-kappa B (NF-κB) is a family of transcription factors that bind the enhancer of the κ light chain gene of B cell immunoglobulin. NF-κB plays an essential role in the activation and development of B cells, and the activation of NF-κB is critical in the inflammation and development of diabetes mellitus. Recently, immunoglobulin-free light chain (FLC) λ was found to be increased in the sera of patients with diabetes mellitus, and the FLC λ and κ/λ ratios are more specific and sensitive markers for the diagnosis of diabetes relative to glycated hemoglobin A1c. Thus, FLCs may be promising biomarkers of inflammation that could relate to the activation of NF-κB. We suggest that NF-κB could be a target for an anti-inflammatory strategy in preventing and treating diabetes when FLCs are modified. FLCs could be a surrogate endpoint in the management of diabetes. In this review, the role of inflammation in the pathogenesis of diabetes, as well as the novel inflammatory biomarkers of FLCs for the management of diabetes, are discussed.

## 1. Introduction

Diabetes mellitus is caused by chronic high glucose levels in the blood as a result of the incapability of β cells in the pancreas to produce adequate insulin or ineffective insulin utilization by cells in the body [1]. There is evidence that virus infection, inflammation and genetic factors play important roles in the pathogenesis of diabetes [2,3,4,5]. Experimental and clinical studies suggest the inflammatory hypothesis, and clinical trials are ongoing to confirm the therapeutic effects targeting inflammation to treat or prevent diabetes [6,7].

The nuclear factor-kappa B (NF-κB) was originally identified as a family of transcription factors that binds the immunoglobulin κ light chain gene enhancer, plays an essential role in the activation and development of B cells, and the activation of NF-κB is critical in the inflammation and development of diabetes mellitus [8,9,10]. Recently, we found that immunoglobulin-free light chains (FLCs) are novel biomarkers of inflammation and found that FLCs are sensitive biomarkers for the diagnosis of inflammatory heart diseases such as heart failure, myocarditis and atrial fibrillation and diabetes [11,12,13]. In this review, the role of inflammation in the pathogenesis of diabetes, and novel inflammatory biomarkers of FLCs for the management of diabetes, are discussed.

## 2. Role of Virus in the Pathogenesis of Diabetes Mellitus

Type 1 diabetes mellitus (T1DM) is believed to be caused by genetic and environmental factors, and viruses are the most well-studied environmental triggers. T1DM is an autoimmune disease in which pancreatic β cells, which produce insulin in normal circumstances, are destroyed. Although multiple genes have been identified to play a role in the development of T1DM, environmental factors may be necessary for its progression to clinical disease [14,15]. Enteroviruses, rotavirus, herpesviruses, and other viruses are thought to be triggers of T1DM [2,3,16,17,18]. Molecular mimicry, direct pancreatic infection, infection-induced changes to the gut mucosa, and interactions between the immune system and infection have been proposed as mechanisms of pathogenesis [14,16,17,19]. Enteroviruses are studied most frequently; however, a growing body of research shows the potential influence of rotavirus on T1DM [20]. Hepatitis C virus (HCV), the most common cause of hepatic failure, is frequently associated with the development of diabetes mellitus, especially type 2 (T2DM) [21].

A recent meta-analysis showed that the odds ratio of risk between non-autoimmune diabetes and virus infections was 10.8 for severe acute respiratory syndrome coronavirus 2 (SARS-CoV-2), 3.6 for HCV, 2.7 for human herpesvirus 8, 2.1 for influenza H1N1 virus, 1.6 for hepatitis B virus, 1.5 for herpes simplex virus 1, 3.5 for cytomegalovirus, 2.9 for Torque teno virus, 2. 6 for parvovirus B19, 0.7 for coxsackie B virus, and 0.2 for hepatitis G virus [22].

### 2.1. SARS-CoV-2

The mechanism of diabetes development in coronavirus disease 2019 (COVID-19) remains to be clarified [23]. COVID-19 affects people with or without diabetes, and hyperglycemia, which is frequently seen in patients with severe COVID-19, is considered as a marker of disease severity [24,25]. A study of COVID-19 patients reported that 22% had a history of diabetes, 21% had newly diagnosed diabetes, and 28% were diagnosed with dysglycemia [25]. A number of studies have reported that new-onset diabetes associated with the presence of COVID-19 was classified as either T1DM or T2DM [23].

Patients with new-onset diabetes have higher levels of inflammatory markers such as C-reactive protein, white blood cell count, and erythrocyte sedimentation rate [25]. A cytokine storm can worsen insulin resistance [26], and neutrophils, d-dimers, and inflammatory biomarkers are higher in individuals with hyperglycemia than in those with normal blood glucose [27]. The proinflammatory cytokines and acute-phase reactants due to COVID-19 may cause inflammation and damage of pancreatic beta cells [28]. A recent study showed that SARS-CoV-2 could infect pancreatic cells, and that the virus entered endocrine islets and exocrine acinar and ductal cells in human pancreatic cultures and postmortem pancreatic tissues from COVID-19 patients [29]. Further studies are needed to investigate the direct effects of SARS-CoV-2 on pancreatic β-cells and other islet cells by experimentation and to assess inflammatory biomarkers in order to understand new-onset COVID-19-related diabetes.

### 2.2. Hepatitis C Virus

Extrahepatic manifestations are frequently seen in chronic HCV infection. About 70% of patients have one or more extrahepatic manifestations over the course of chronic HCV infection, which are often the first and only clinical signs and symptoms of infection. A causal association between extrahepatic manifestations such as cardiovascular disease, insulin resistance, T2DM, mixed cryoglobulinemia, non-Hodgkin lymphoma, neurological and psychiatric diseases, and rheumatic disease and HCV infection has been supported by experimental and clinical evidence [21,30].

Meta-analyses have shown an approximately 3.5- to 3.6-fold increase in HCV infection risk in individuals with T2D [22,31], and HCV infection seems to be strongly associated with non-autoimmune diabetes. An analysis reported an approximately 1.7-fold increase in T2DM risk in HCV infected individuals compared with non-infected individuals [32,33], and HCV infection is associated with an increased risk of T2DM independent of the severity of the associated liver disease. Patients with chronic HCV have higher insulin resistance compared with body mass index–matched controls, and viral eradication improves global, hepatic, and adipose tissue insulin sensitivity [34], suggesting that HCV infection precedes non-autoimmune diabetes.

HCV replicates in pancreatic cells and affects insulin signaling pathways through its structural and non-structural proteins [35]. The indirect mechanisms of insulin resistance involve HCV-induced oxidative stress, the release of inflammatory cytokines, and the upregulation of gluconeogenic genes such as glucose 6 phosphatase and phosphoenolpyruvate carboxy kinase 2 [36]. Recent studies have shown that clearance of HCV by direct-acting antiviral agents (DAAs) leads to improvement or regression of insulin resistance, improves control of glucose homeostasis in patients both with and without T2DM, and reduces the incidence of T2DM [37,38]. A prospective study of over 2400 HCV patients demonstrated an 81% reduction in the risk of developing T2DM in those who were treated with DAAs compared to those who were untreated [37]. These studies suggest that HCV plays a central role in the increased risk of developing insulin resistance and T2DM, and that eliminating the HCV can reverse insulin resistance and prevent the development of T2DM [39].

### 2.3. A New Concept of Pathogenesis of HCV-Induced Diseases

HCV infection is frequently associated with heart diseases such as myocarditis, dilated cardiomyopathy, arrhythmogenic right ventricular cardiomyopathy and hypertrophic cardiomyopathy. Various arrhythmias, conduction disturbances and QT prolongation were also associated with HCV infection [21,40,41,42,43]. We found that CD68-positive monocytes/macrophages were a primary target of HCV infection [44]. HCV-core antibodies stained mostly mononuclear cells in various body organs such as the liver, heart, kidney and bone marrow, but not hepatocytes or myocytes. Antibodies against the NS4 protein stained the mononuclear cells of peripheral blood and various tissues, confirming that HCV replicates in the mononuclear cells [44].

The presence of multiple extrahepatic organ involvement could be explained by the effect of HCV-infected monocytes/macrophages by immune escape and viral modulation of host immune responses. The virus may also spread through the lymphatic system, where it reaches the peripheral lymph nodes, which may cause immune cell infection prior to recirculation. Thus, HCV may cause diabetes by inflammation in the pancreas induced by monocytes/macrophages infected with HCV.

The major human histocompatibility complex (MHC) is located on the short arm of chromosome 6 and codes for several cell surface proteins involved in immune function, such as complement system components. There are marked differences in the MHC-related disease susceptibility for HCV-associated cardiomyopathies, which suggests that HCV-associated cardiomyopathies are controlled by different pathogenic mechanisms [45,46]. Therefore, HCV-induced diabetes might associate with different MHCs [3].

## 3. Role of Inflammation in the Pathogenesis of Diabetes Mellitus

Diet influences inflammation. Orally absorbed advanced glycation and lipoxidation end-products that are formed during the processing of foods are linked to overnutrition and hence obesity and inflammation. Furthermore, high-glycemic-load foods, such as isolated sugars and refined grains can cause increased oxidative stress that activates inflammatory genes [47]. Physical inactivity can increase the risk for diabetes because it is linked to obesity, and excessive visceral adipose tissue is a significant trigger of inflammation [47].

### 3.1. Inflammatory Cytokines

Circulating levels of acute-phase proteins are elevated in diabetes, such as serum amyloid A, C-reactive protein (CRP), fibrinogen, haptoglobin, plasminogen activator inhibitor, sialic acid, interleukin (IL)-1*β*, IL-1 receptor antagonist (IL-1Ra), IL-6 and tumor necrosis factor (TNF)-*α* [48,49,50,51]. Elevated circulating CRP, IL-1*β*, IL-1Ra and IL-6 are predictive markers for the development of T2DM [49,52,53,54,55]. The production of TNF-*α* is increased by adipose tissues during obesity, and insulin sensitivity is improved by a TNF-*α* antagonist [56]. Macrophages and other immune cells exist in adipose tissues and may release TNF-*α*, IL-1*β*, IL-6 and IL-33 [57,58,59]. It is now well-established that tissue inflammation plays a critical role in insulin resistance [6,7].

Inflammation may play an important role in defective insulin action and insulin secretion. Increased cytokine expressions and immune cell infiltration of pro-inflammatory macrophages are seen in pancreatic islets of patients with T2DM [60,61]. This chronic inflammatory process is associated with fibrosis and amyloid deposits, which are observed in the islets of most patients with T2DM [7].

### 3.2. Nuclear Factor-Kappa B (NF-κB)

Nuclear factor-kappa B (NF-κB) is a key molecule in the pathogenesis of diabetes. The NF-κB pathway is activated by genotoxic, oxidative and inflammatory stress, and regulates the expression of cytokines, growth factors and genes that regulate apoptosis, cell-cycle progression and inflammation [8]. Pharmacologic and genetic suppression implicated that NF-κB activation causes insulin resistance and glucose metabolism [9]. Upregulation of NF-κB signaling in hepatocytes results in a T2DM [10], and innate immune activation and inflammatory response that may underlie T2DM [62]. Therefore, NF-κB activation in numerous tissues, including adipose tissue, pancreas and liver, contributes to the pathogenesis of T2DM.

## 4. Novel Biomarkers of Inflammation: Immunoglobulin-Free Light Chains (FLCs)

### 4.1. FLCs as Novel Biomarkers of Chronic Inflammation

NF-κB was originally identified as a family of transcription factors that binds the immunoglobulin κ light chain gene enhancer. FLCs are synthesized de novo and secreted into circulation by B cells. FLCs emerge as an excess byproduct of antibody synthesis by B cells; elevated FLCs have been proposed to be a biomarker of B cell activity in many inflammatory and autoimmune conditions [63]. Polyclonal FLCs are a predictor of mortality in the general population, measured by the sum of κ and λ concentrations [64]. Increased FLCκ, and the higher κ/λ ratio, occurred more in rheumatic disease than in healthy blood donors [65]. FLCs in inflammatory and autoimmune diseases correlate with disease activity, suggesting their role as potential therapeutic targets in such conditions.

As discussed above, HCV infection can induce insulin resistance and cause diabetes [20,21,22,23,24,25,26,27,28,29,30,31,32,33,34,35,36,37,38]. High concentrations of FLC κ have been observed in HCV-positive patients, and an alteration in the κ/λ ratio is positively correlated with an increasing HCV-related lymphoproliferative disorder severity [66]. Furthermore, it has been suggested that the κ/λ ratio may be useful in the evaluation of therapeutic efficacy [67].

### 4.2. FLCs as Markers of Heart Failure and Myocarditis

We found that FLCs were increased in a mouse model of heart failure due to viral myocarditis [68]. Recently, we conducted additional research with patients in heart failure, and we observed that circulating FLC λ were increased while the κ/λ ratio was decreased in sera from patients with heart failure resulting from myocarditis, as compared to a group of healthy controls. These findings demonstrated that the FLC λ and κ/λ ratio together showed good diagnostic potential for the identification of myocarditis. In addition, the FLC κ/λ ratio could also be used as an independent prognostic factor for overall patient survival [11].

As shown in our previous studies, HCV infection has often been associated with myocarditis [21,40,41,42,43,44,45]. In our study on FLCs using sera from the U.S. Multicenter Myocarditis Treatment Trial, myocardial injury was more severe in patients with HCV infection than in non-infected patients. The level of FLC κ was lower, FLC λ was higher, and the κ/λ ratio decreased in patients with myocarditis, both with and without biopsy-confirmation according to the Dallas criteria, as compared to normal volunteers. These changes were more prominent in patients with HCV infection, as compared to those without infection. HCV infection may enhance the production of FLC λ while decreasing FLC κ [69,70]. Although the mechanisms of these changes require clarification, the detection of FLCs might be helpful for the diagnosis of myocarditis with heart failure and also be useful in differentiating patients with HCV infection from those without infection [69,70]. In heart failure patients, LV end-diastolic and end-systolic diameters, pulmonary arterial pressure, and N-terminal pro-brain natriuretic peptide correlated positively with FLC λ and negatively with the κ/λ ratio. Left ventricular ejection fraction was also negatively correlated with the κ/λ ratio [70].

### 4.3. FLCs and COVID-19 and Heart Diseases

The recent review of 316 cases of postmortem examination of COVID-19 patients demonstrated that cardiac abnormalities, either on gross pathology or histology, were identified in almost all cases. Most autopsies demonstrated chronic cardiac pathologies such as hypertrophy (27%), fibrosis (23%), amyloidosis (4%), cardiac dilatation (20%), acute ischemia (8%), intracardiac thrombi (2.5%), pericardial effusion (2.5%), and myocarditis (1.5%). SARS-CoV-2 was detected within the myocardium of 47% of studied hearts [71]. However, the Dallas criteria was satisfied in only five of these cases. In an additional 35 cases, minimal lymphocytic or mononuclear infiltration was reported, and they did not satisfy the Dallas criteria for myocarditis. Lymphocytic infiltration was scarce but could be detected in the pericardium, myocardium, epicardium, or endothelium. Therefore, cellular infiltration may be rare in COVID-19 myocarditis and, therefore, the Dallas criteria may not be accurate in the diagnosis of COVID-19 myocarditis, as it is the same in the case of HCV myocarditis [21,69].

An increase in blood troponin levels in COVID-19 is an indicator of myocardial damage. Several studies have documented a strong association between COVID-19 progression and elevated blood troponin. Reports from China found that elevated circulating cardiac troponin was present in 7–28% of COVID-19 patients, suggesting the existence of myocardial injury or myocarditis [72,73]. In hospitalized patients with COVID-19, mortality in the elevated-blood-troponin group was 51.2–59.6%, a range markedly higher than in the 4.5–8.9% in the normal-blood-troponin group [74].

We have studied how frequently myocardial injury or myocarditis occurs in COVID-19 patients [75]. Troponin T was positive in 63% of patients, NT-proBNP was elevated in 68% of patients, and elevated creatine kinase was noted in 43% of patients at admission. NT-proBNP showed a significant correlation with the length of hospital management and the severity of pulmonary CT findings. In addition, the existence of enhanced inflammatory biomarkers such as CRP and ferritin suggested that myocardial injury may be caused by inflammatory myocardial processes. D-dimer was also elevated frequently, suggesting that coagulation abnormality occurs frequently in COVID-19 patients [75]. Thus, COVID-19 has been frequently associated with myocardial injury, suggesting that SARS-CoV-2 causes myocarditis.

We also measured FLCs and IL-6 in COVID-19 patients. FLC κ and λ was elevated in 73% and 80% of patients, respectively, and the frequency of the elevated levels was higher than those of troponin T, NT-proBNP, creatine kinase, and IL-6. IL-6 has been frequently measured in COVID-19 patients, but elevated levels of IL-6 were less frequent, as compared to other parameters [69,75].

### 4.4. FLCs as Markers of Atrial Fibrillation

Atrial fibrillation is the most common arrhythmia, which is an important cause of stroke. Diabetes is a risk factor for the development of atrial fibrillation. Diabetes in patients with atrial fibrillation is associated with increased cardiovascular and cerebrovascular mortality [76]. The pathogenesis of diabetes-related atrial fibrillation remains to be clarified, but may be related to structural, electrical, electromechanical, and autonomic remodeling.

Abnormal atrial histology compatible with a diagnosis of myocarditis was uniformly found in patients with lone atrial fibrillation. Patients with atrial fibrillation exhibited a higher concentration of cytokines, higher NF-κB activity and more severe lymphocyte infiltration than those in sinus rhythm. These observations imply local inflammatory responses in the atria in atrial fibrillation [12]. The concentrations of circulating FLC κ and λ in patients with lone atrial fibrillation were significantly different from the healthy group. The mechanism by which FLCs cause atrial fibrillation remains to be clarified. However, the inflammation associated with FLCs directly induces atrial fibrillation. Moreover, FLCs might cause a change in membrane fluidity, which, in turn, could alter ion channel function [12].

### 4.5. FLCs as Biomarkers of Diabetes

Since we found that FLCs could be biomarkers of NF-κB, immune responses and inflammation, FLCs were measured in the patients with T2DM. Circulating levels of FLC λ were higher, and the κ/λ ratio was lower in patients with T2DM than in controls (Figure 1) [13].

A statistical analysis showed that the area under the receiver operating curve (ROC-AUC) of the FLC λ and κ/λ ratio was significantly larger than glycated hemoglobin (HbA1c) [13]. The diagnostic ability for distinguishing between T2DM and controls had a sensitivity of 0.96, a specificity of 1, a positive predictive value of 1 and a negative predictive value of 0.96, with an optimal cutoff value of 1.3 for the FLC κ/λ ratio,. The odds ratio was 0.000018. The ROC-AUC, sensitivity, and specificity for HbA1c were 0.95, 0.86 and 0.94, respectively, on the cutoff value of 6.2% (Figure 2 and Figure 3).

In our preliminary study, urine FLC λ and the κ/λ ratio were well correlated with sera, suggesting that urine FLCs could be a suitable and non-invasive biomarker of diabetes (unpublished observation). Since HbA_1c_ cannot be measured in urine, FLCs would be more beneficial biomarkers of diabetes than HbA1c. Since FLCs are a marker of inflammation/immune activation, their presence in diabetes confirms the inflammatory /immune character of the disease.

Since NF-κB activation is a critical mechanism of the inflammatory cascade in developing T2DM as discussed above [8,9,10], it is interesting that FLC λ and κ/λ ratio are more specific and sensitive markers for the diagnosis for T2DM than HbA1c. Therefore, FLCs represent promising potential biomarkers of inflammation that may reflect the activation of NF-κB.

Recently, we also found that FLC λ was higher, and the κ/λ ratio was lower in patients with T1DM, as seen in those with T2DM (unpublished observation). The reason why the specific activation of FLC λ occurred is unknown. B lymphocytes and plasma cells, which produce FLC λ, may be specifically activated in diabetes [13]. Another possibility is that FLC κ and λ are differently regulated because NF-κB may not exercise control of the production of FLC κ and λ in the same manner [13]. NF-κB could be a target for new types of anti-inflammatory therapy for diabetes when FLCs are changed and could be a surrogate endpoint in the management of diabetes.

## 5. Targeting Inflammation for the Management of Diabetes

Several therapeutic approaches or pharmacologic agents used for diabetes are reported to have anti-inflammatory properties in addition to their major mechanisms of action. Conversely, some anti-inflammatory approaches may affect glucose metabolism and cardiovascular health. It is suggested that targeting the inflammation may differentially affect hyperglycemia and atherothrombosis. Clarifying the underlying pathogenetic mechanisms may contribute to the development of effective new therapies for the optimal management of both metabolic and atherothrombotic disease states [6,7].

### 5.1. Metformin

Cytokines and chemokines play important roles in inflammation, and some of them are therapeutic targets for attenuating chronic inflammatory diseases [77,78]. In a large-scale treatment trial of newly diagnosed diabetic patients, metformin decreased the neutrophil-to-lymphocyte ratio, a marker of systemic inflammation. Metformin also inhibited circulating cytokines and chemokines in a non-diabetic heart failure trial. These findings show that metformin has anti-inflammatory effects in both diabetic and non-diabetic patients [79]. Metformin attenuates the production of IL-6 and TNF-α induced by lipopolysaccharide (LPS) and reduces the activation of NF-κB induced by TNF-α. NF-κB inhibition by metformin also reduces IL-1β production [80]. Metformin was shown to inhibit LPS-stimulated chemokine expression by activating AMP-activated protein kinase (AMPK), and to inhibit the phosphorylation of I-κBα and p65 in a macrophage cell line [78]. Metformin also attenuated LPS-stimulated acute lung injury by activating AMPK; reducing inflammatory cytokine, neutrophil, and macrophage infiltration; and reducing myeloperoxidase activity [81]. Metformin therapy reduced acute phase serum amyloid A, a pro-inflammatory adipokine that is upregulated in patients with obesity and insulin resistance [82]. The anti-inflammatory actions of metformin seem to be independent of glycemia and are most prominent in immune cells and vascular tissues [6].

### 5.2. Dipeptidyl Peptidase-4 Inhibitors

Dipeptidyl peptidase-4 (DPP-4) is a transmembrane glycoprotein known as CD26, expressed on T lymphocytes, macrophages and endothelial cells, and regulates the actions of chemokines and cytokines involved in T cell activation. DPP-4 inhibitors suppress the actions of NLRP3 inflammasomes, TLR4 and IL-1β in macrophages [83]. Sitagliptin and other DPP-4 inhibitors reduce the expression or activity of TNF-α, jun amino terminal kinase (JNK)1, Toll-like receptor (TLR) 2, TLR4, β subunit of IκB kinase and the chemokine receptor CCR2 [84].

### 5.3. The Glucagon-Like Peptide 1 Receptor Agonists

The glucagon-like peptide 1 (GLP-1) receptor agonists reduce circulating inflammatory biomarkers even in the absence of substantial weight loss. Markers of inflammation, are reduced including reactive oxygen species, NF-κB activity, the expression of mRNAs of IL-1*β*, TNF-α, JNK1, TLR2, TLR4 and SOCS-3 in mononuclear cells, and circulating concentrations of IL-6, monocyte chemoattractant protein-1, matrix metalloproteinase-9, and serum amyloid A [85,86].

### 5.4. SGLT2 Inhibitors

SGLT2 inhibitors improve cardiovascular and renal outcomes in large cardiovascular outcome trials in patients with diabetes. SGLT2 inhibitors reduce adipose tissue-mediated inflammation and pro-inflammatory cytokine production [87,88]. An SGLT2 inhibitor, canagliflozin, was reported to decrease circulating levels of IL-6, TNF receptor 1, fibronectin 1 and matrix metalloproteinase 7, and contributes to improving molecular processes related to inflammation, extracellular matrix turnover and fibrosis [89]. Empagliflozin may contribute to cardiovascular benefits in heart failure by repleting AMP kinase activation-mediated energy and reducing inflammation [90].

### 5.5. Anti-IL-1 Agents

Anakinra (recombinant human IL-1 receptor antagonist) improved glycemia, reduced CRP levels and improved *β*-cell secretory function [91]. The CANTOS study demonstrated that anti–IL-1*β* antibody (canakinumab) treatment lowered cardiovascular events over placebo [92]. IL-1*β* antagonism significantly decreased HBA1c in a subanalysis on metabolic endpoints [93]. A T2DM meta-analysis, following the CANTOS study, demonstrated a substantial reduction in HbA1c [94].

Therapeutic approaches to reduce inflammation may include weight-reducing diets and lifestyles, pharmacologic or surgical approaches to weight management, statin therapy and antidiabetic drugs. Serial measurements of FLCs in these interventions may be helpful in the evaluation of their therapeutic efficacy as anti-inflammatory interventions. The determination of FLCs seems suitable as an initial health screening in the general population. When the abnormalities of FLCs are found, secondary tests such as HbA1c would be performed and followed up for diabetes.

Figure 4 summarizes the risk factors, inflammation, FLCs and anti-inflammatory therapy for diabetes.

## 6. Conclusions

Virus infection and inflammation are important factors in the pathogenesis of diabetes mellitus. Enteroviruses are studied most frequently; however, a growing body of research shows the potential influence of HCV and SARS-CoV-2 infections in the pathogenesis of T1DM and T2DM. Circulating FLCs are specific and sensitive diagnostic markers for diabetes mellitus. They may represent promising potential biomarkers of inflammation, which may reflect activation of NF-κB. NF-κB could be a target for new types of anti-inflammatory prevention and treatment for diabetes when FLCs are changed. FLCs could be a surrogate endpoint in the management of diabetes. Anti-inflammatory approaches may be promising for the prevention and treatment of diabetes mellitus. Clarifying the underlying pathogenetic mechanisms may contribute to the development of effective new therapies for optimal management of both metabolic and cardiovascular diseases.

## Figures and Tables

**Figure 1 biomedicines-10-00666-f001:**
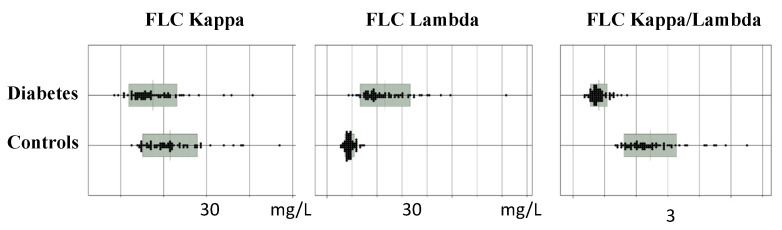
Immunoglobulin-free light chains (FLCs) in patients with type 2 diabetes and healthy controls (Adapted from [13]).

**Figure 2 biomedicines-10-00666-f002:**
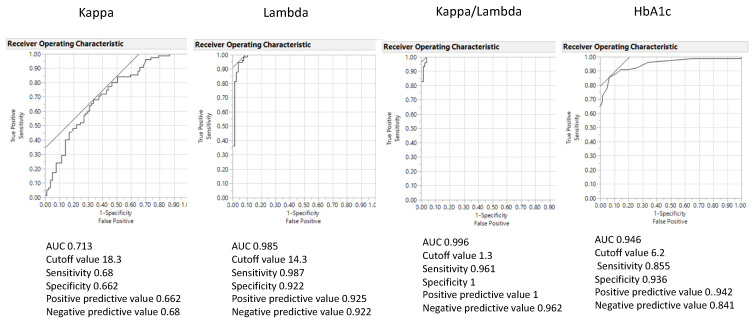
The area under the receiver operating curve (ROC-AUC) of the FLC κ, λ and κ/λ ratio and glycated hemoglobin (HbA1c). ROC-AUC of the FLC κ/λ ratio showed the largest compared with other FLC variables (Adapted from [13]).

**Figure 3 biomedicines-10-00666-f003:**
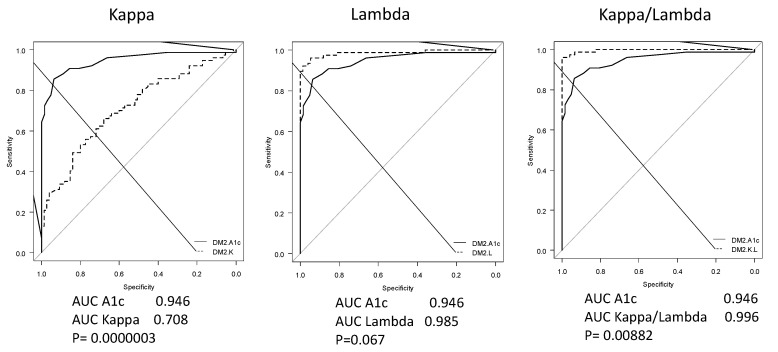
Comparisons of ROC-AUC between FLC variables and HbA1c. The ROC-AUC of the FLC κ/λ ratio was larger than that of HbA1c (Adapted from [13]).

**Figure 4 biomedicines-10-00666-f004:**
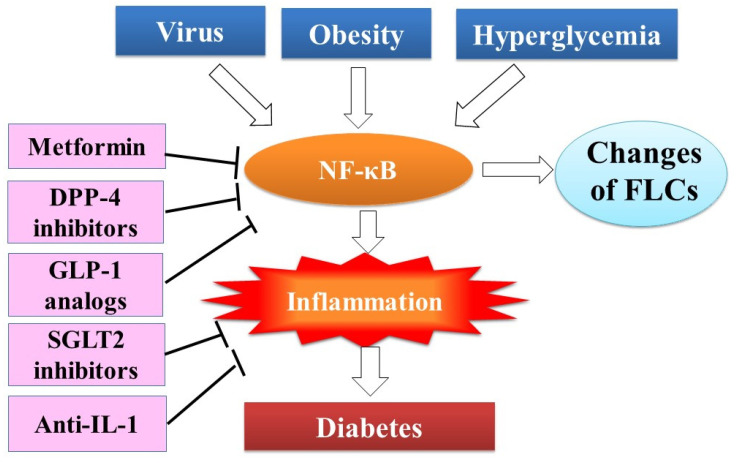
FLCs as inflammatory biomarkers of diabetes. Risk factors of diabetes such as viral infection, hyperglycemia and obesity activate nuclear factor kappa B (NF-κB), which regulates transcription of immunoglobulin-free light chains in the immunoglobulin-producing B cells and plasma cells and production of many inflammatory molecules, leading to inflammation. Thus, FLCs were proposed to be biomarkers of NF-κB activation and inflammation. Metformin, DPP-4 inhibitors and GLP-1 receptor agonists inhibit NF-κB activation and inflammation, and SGLT2 inhibitors and anti-IL-1 therapy inhibit inflammation.

## Data Availability

Not applicable.

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
