# Peer review of "Novel Biomarkers of Inflammation for the Management of Diabetes: Immunoglobulin-Free Light Chains"

_biomedicines, 2022, doi:10.3390/biomedicines10030666_

Round 1

Reviewer 1 Report

In the review entitled “Novel Biomarkers of Inflammation for the Management of Diabetes: Immunoglobulin-Free Light Chains” Akira Matsumori discussed the implication of viruses in activating inflammation in several deseases. In particular, the Author focusing on the role of immunoglobulin-free light chain (FLC) λ in the pathogenesis of diabetes. FLC was found to be increased in the sera of patients with diabetes mellitus. Morover, FLC λ and κ/λ ratios are showed more specific and sensitive markers for the diagnosis of diabetes relative to hemoglobin A1c. Due to the possible implication of FLC in the activation of NF-kB, FLCs may be promising inflammatory biomarkers for diabetes managment.

The review is well written, rationale, background and figures are clear and exhaustive. Some very minor concerns are to be assessed to the Author:

  • Row 26: Add reference Yoon JW.; Austin M.; Onodera T.; Notkins AL. Isolation of a virus from the pancreas of a child with diabetic ketoacidosis. N Engl J. 1979, 300 (21), 1173-9.
  • Row 262: Author should replace “figure” with “Figure” to be consistent with the rest of typing.
  • Reference 35,36,37 should be in black rather than in red.

Author Response

Response:

Raw26: I added the reference as suggested.

Raw 262: "figure" was replaced with Figure.

References 35-37 were changed to black.

Reviewer 2 Report

The author describes the role of immunoglobulin-free light chain (FLC) as a surrogate biomarker for diabetes. It is a comprehensive review and nicely described and presented data to convince how this FLC could be a potential marker for diabetes management. Well written and nicely presented. I have no additional comments or concerns about the manuscript.

Author Response

Thank you for your review.

Reviewer 3 Report

Dear Editor,

Akira Matsumori's manuscript titled "Novel Biomarkers of Inflammation for the Management of Diabetes: Immunoglobulin-Free Light Chains" is interesting and well-structured review. However, some points need some clarification:

  1. In the introduction it should be emphasized that without an absolute or relative defect in insulin secretion, diabetes mellitus cannot onset.
  2. The potential role of inflammation in the pathogenesis of diabetes mellitus is very well explained. However, it should be also outlined that a pro-inflammatory state can be secondary to a high-calorie diet rich in saturated fatty acids or secondary to other lifestyle defects etc.
  3. Part 5 Targeting Inflammation for the Management of Diabetes by the Novel FLC Biomarker should be improved since there is a significant difference from the other sections of the manuscript.
  4. Figure 4 is too simple and need to be improved.

Author Response

Responses:

I revised the introduction as suggested, and added reference 1.

"Diabetes is caused by chronic high glucose levels in the blood as a result of the incapability of β cells in the pancreas to produce adequate insulin or ineffective insulin utilization by cells in the body [1]."

Alberti, K. G., Zimmet PZ. Definition, diagnosis and classification of diabetes mellitus and its complications. Part 1: diagnosis and classification of diabetes mellitus provisional report of a WHO consultation. Diabet. Med. 1998; 15(7): 539-53.

2. I discussed on influence on diet and exercise in row 130 as suggested, and added reference #37.

"Diet influences inflammation. Orally absorbed advanced glycation and lipoxidation end-products that are formed during the processing of foods are linked to overnutrition and hence obesity and inflammation. Furthermore, high-glycemic-load foods, such as isolated sugars and refined grains can cause increased oxidative stress that activates inflammatory genes [37]. Physical inactivity can increase risk for diabetes because it is linked to obesity, and excessive visceral adipose tissue is a significant trigger of inflammation [37]."

37. Furman. D., Campisi, J., Verdin, E., Carrera-Bastos, P., Targ, S., Claudio Franceschi, C., Lu Ferrucci, L., Gilroy, D. W., Fasano, A., Miller, G. W., Miller, A.H., Mantovani, A., Weyand, C. M., Barzilai, N., Goronzy, J. J., Rando, T.A., Effros, R. B., Lucia, A., Kleinstreuer, N., Slavich, G. M. Chronic inflammation in the etiology of disease across the life span. Nat. Med. 2019. 25(12): 1822–1832.

3. Subtitle of Part 5 has been modified as "Targeting Inflammation for the Management of Diabetes" as suggested.

4. Figure 4 has been revised as suggested.